# Climate change and intensive land use reduce soil animal biomass via dissimilar pathways

Rui Yin[1,2,3]*, Julia Siebert[2,3], Nico Eisenhauer[2,3†], Martin Schädler[1,2†]

[1]Department of Community Ecology, Helmholtz-Centre for Environmental Research-UFZ, Halle, Germany; [2]German Centre for Integrative Biodiversity Research (iDiv), Halle-Jena-Leipzig, Leipzig, Germany; [3]Institute for Biology, Leipzig University, Leipzig, Germany

**Abstract** Global change drivers, such as climate change and land use, may profoundly influence body size, density, and biomass of soil organisms. However, it is still unclear how these concurrent drivers interact in affecting ecological communities. Here, we present the results of an experimental field study assessing the interactive effects of climate change and land-use intensification on body size, density, and biomass of soil microarthropods. We found that the projected climate change and intensive land use decreased their total biomass. Strikingly, this reduction was realized via two dissimilar pathways: climate change reduced mean body size and intensive land use decreased density. These findings highlight that two of the most pervasive global change drivers operate via different pathways when decreasing soil animal biomass. These shifts in soil communities may threaten essential ecosystem functions like organic matter turnover and nutrient cycling in future ecosystems.

**\*For correspondence:**
rui.yin@ufz.de

[†]These authors contributed equally to this work

**Competing interests:** The authors declare that no competing interests exist.

## Introduction

Anthropogenic environmental changes are altering ecological communities and ecosystem functions (*Chapin et al., 2000*; *Sala et al., 2000*). As one of the most pervasive drivers, climate change tends to decrease invertebrate density in different ecosystems (*Barry et al., 1995*; *Kardol et al., 2011*; *Lister and Garcia, 2018*). Moreover, climate change may shift the functioning and evolutionary adaptations of communities (*Briones et al., 2009*; *Hoffmann and Sgrò, 2011*). For instance, it has substantial influences on population-level phenotypic trait expression of organisms, such as shifts in morphology, that is, body size and shape (*Gardner et al., 2011*).

As warmer conditions increase individual metabolism (*Scheffers et al., 2016*) and development rates (*Atkinson et al., 2003*), many groups of organisms (e.g., plants, fish, ectotherms, birds, and mammals) have already been reported to shrink their body size in response to warming (*Sheridan and Bickford, 2011*). These shifts in body size may result in a wide range of implications, e.g., biomass loss, including negative effects on the structure and dynamics of ecological networks (*Woodward et al., 2005*).

Precipitation regimes drive shifts the structure (abundance and diversity) of soil microarthropod communities across different terrestrial ecosystems (*Sylvain et al., 2014*). A meta-analysis showed that the positive effect size of increased precipitation intensifies with experimental time (*Blankinship et al., 2011*). However, precipitation may also disturb microarthropod communities, resulting in abundant loss of saturated conditions (*Turnbull and Lindo, 2015*). By contrast, droughts generally reduce the abundance (*Landesman et al., 2011*) and change the morphology of soil animals (*Andriuzzi et al., 2020*). For example, soil animals may decrease in body size and biomass to reduce their physiological requirements in response to drought events. Corresponding

morphophysiological shifts may cause potential alterations in ecosystem functions, such as decomposition and nutrient cycling (*Eisenhauer et al., 2012*; *Wall et al., 2008*). However, climate change is often a combination of both warming and altered precipitation (e.g., ranging from periods of prolonged drought to extreme precipitation events), which may reshape biocenosis in terrestrial ecosystems. Yet organisms with small body size in general, and soil-living organisms in particular, have not received much attention in this combined context (*Thakur et al., 2018*; *Torode et al., 2016*).

Besides climate change, soil systems further strongly suffer from land-use intensification, for example, conversion of grasslands into croplands and intensified management practices (*Foley et al., 2011*). Such practices include tillage, mowing, livestock grazing, heavy machine employment, as well as herbicide, pesticide, and fertilizer application, all of which may profoundly endanger soil communities, as well as the functions and services they provide (*Giller et al., 1997*; *McLaughlin and Mineau, 1995*; *Newbold et al., 2015*; *Tsiafouli et al., 2015*). Accordingly, land-use change is considered as the major global threat for biodiversity (*Sala et al., 2000*), and this view also holds for soil ecosystems (*Smith et al., 2016*). It has been shown that grasslands are more suitable habitats for soil microarthropod (diversity and abundance) conservation, compared with croplands (*Menta et al., 2011*). In cropland systems, benefits of organic farming (e.g., mechanical weed control, organic fertilization, non-stained seeds, and restricted use of pesticides) on soil microarthropod communities and associated ecosystem functions have been widely confirmed. As a result, organic farming has been considered to represent a potential approach to reduce the environmental impact of agriculture (*Domínguez et al., 2014*; *House and Parmelee, 1985*).

In grassland systems, intensively-managed grasslands (i.e., frequent mowing and fertilization) appear to be incompatible with maintaining a high level of biodiversity and a complex community structure (*Plantureux et al., 2005*), while the effects of grazing on soil biota are controversial and may be species specific and context dependent (*Qi et al., 2011*). Generally, the trampling and feeding behavior of livestock often has detrimental effects on both aboveground and belowground communities; however, livestock manure may increase resource availability for soil food webs, e.g., increased biological activity, abundance, and diversity (*Andrés et al., 2016*; *Epelde et al., 2017*).

In this context, land-use intensification is deemed to decrease the abundance and biodiversity of soil organisms (*Bardgett and van der Putten, 2014*; *Flynn et al., 2009*; *Postma-Blaauw et al., 2010*), consequently threatening the functioning of soils and the ecosystem services that they deliver, like soil fertility and nutrient dynamics (*Beare et al., 1992*; *de Vries et al., 2013*; *Yin et al., 2019*), which may be fed back to primary production (*Cardinale et al., 2004*). Furthermore, land-use intensification can have implications for trait diversity and functional composition in aboveground and belowground arthropod communities (*Birkhofer et al., 2017*; *Yin et al., 2020*). For example, frequent perturbations in intensive land use may select for soil microarthropods with particular life-history traits, such as *r*-strategists with high reproduction rates and small body size.

Taken together, both climate change and land-use intensification may decrease the biomass of soil microarthropods by decreasing their mean body size and density. Such changes in soil communities would be alarming given that many important ecosystem functions are determined by the biomass of soil organisms (*Höfer et al., 2001*; *Horwath, 1984*; *Petersen and Luxton, 1982*). Moreover, effects of climate change on soil communities and their functions can be dependent on environmental contexts, such as different land-use regimes (*de Vries et al., 2012*; *Foley et al., 2005*; *Walter et al., 2013*). That is, intensively-managed land characterized by higher levels of disturbance and lower biodiversity may be more vulnerable to climate change (*Isbell et al., 2017*); while extensively-managed land, with less disturbance and higher biodiversity potentially mitigates these negative effects of climate change (*Oliver et al., 2016*). Therefore, disentangling the pathways by which these main environmental change drivers contribute to changes in the biomass of soil organisms and identifying potential interactive effects is crucial to better understand how ecosystem functions and services may be affected and could be maintained in the future.

To address this critical knowledge gap, we tested potential interactive effects of climate and land use on body size, density, and biomass of soil microarthropods. This study was conducted at the Global Change Experimental Facility (GCEF) in Central Germany, where climatic conditions are manipulated following a future scenario for the years 2070–2100 with increased temperature (ambient vs. ~0.6°C warming) and altered precipitation patterns (i.e., 20% reduction in summer, and 10% addition in spring and autumn, respectively) across five different land-use regimes (i.e., two croplands and three grasslands differing in management intensity). We used data of multiple sampling

campaigns to test how climate, land use, and the interaction of these two could affect the total biomass of soil microarthropods. We hypothesized that (1) climate change, intensive land-use type (i.e., croplands), and intensified management practices (e.g., conventional farming in croplands, or intensive management in meadows) will decrease the body size and density of soil microarthropods, which then causes a reduction in total microarthropod biomass. Moreover, (2) we expected to find synergistic effects of these two environmental change drivers as negative climate change effects may be particularly strong in intensively-used land; by contrast, in extensively-used land, these negative effects can be diminished.

## Results

### Climate change reduces the body size of soil microarthropods

Climate change significantly reduced the body size of microarthropods by ~10% (*Figure 1A*; *Table 1A*), which was driven by multiple taxa. Specifically, the body size of Oribatida, Mesostigmata, and Sminthuridae significantly decreased under the future climate scenario (*Figure 1C–D*; *Table 1A*). However, land-use treatments did not significantly affect the body size of microarthropods and their taxa (*Table 1A*).

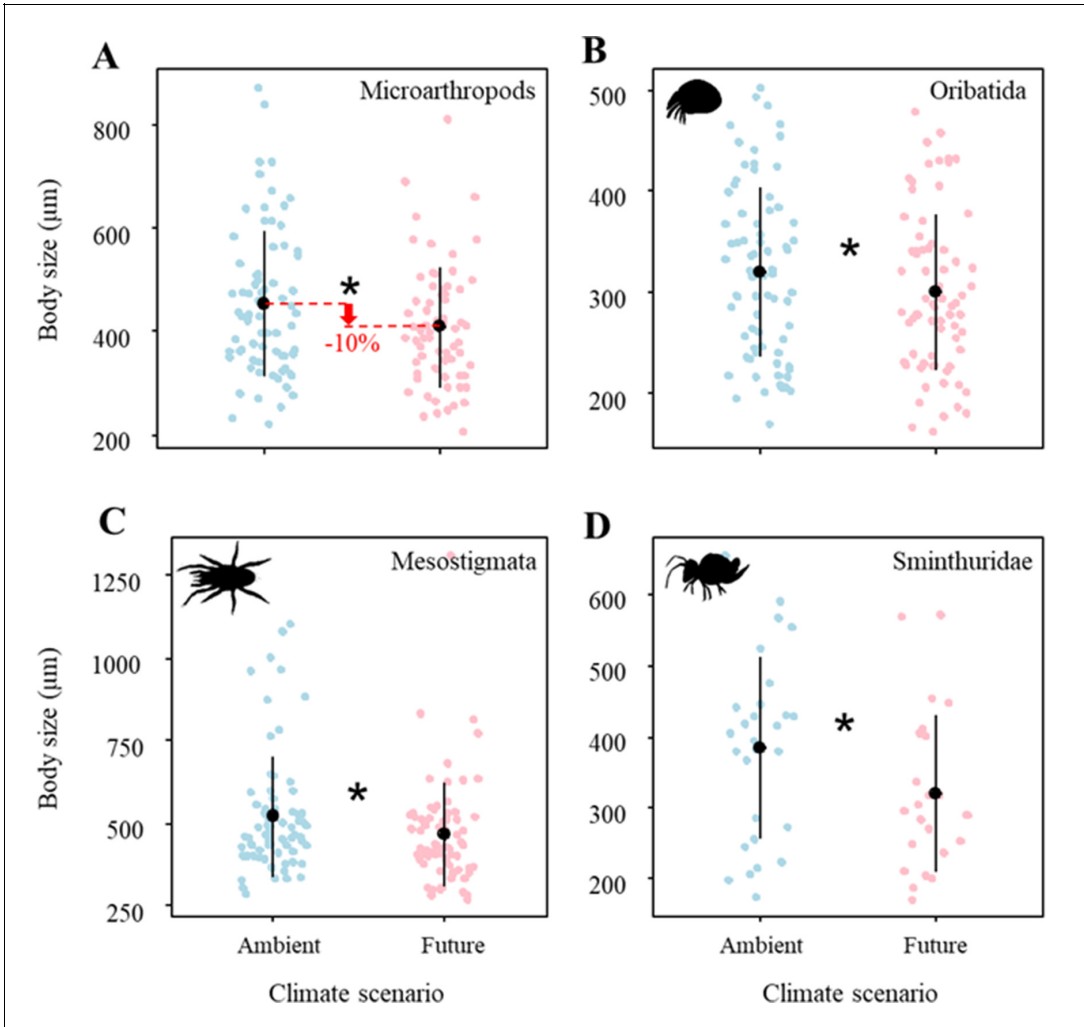

**Figure 1.** Effects of climate on the body size of (A) Microarthropods, (B) Oribatida, (C) Mesostigmata, and (D) Sminthuridae. The predicted mean ± SD of body size for the ambient climate scenario (with all raw data as blue points), and future climate scenario (with all raw data as red points). * denotes significant (P < 0.05) differences between climate scenarios based on post-hoc Tukey's HSD tests.

**Table 1.** Results (*F*-values) from generalized linear mixed models testing the effects of climate, land use, season and their interactions on (**A**) body size, (**B**) density and (**C**) biomass of soil microarthropods, Acari (including its order, i.e., Orib_: Oribatida; Meso_: Mesostigmata; Pros_: Prostigmata), and Collembola (including its family, i.e., Isot_: Isotomidae; Ento_: Entomobryidae; Smin_: Sminthuridae; Kati_: Katiannidae; Hypo_: Hypogastruridae; Onyc_: Onychiuridae).

Significant effects are indicated in bold font, with †=P < 0.1, *=P < 0.05, **=P < 0.01, ***=P < 0.001.

| | Independent variable | Df | Micro-arthropod | Acari | Orib_ | Meso_ | Pros_ | Collem-bola | Isot_ | Ento_ | Smin_ | Kati_ | Hypo_ | Onyc_ |
|---|---|---|---|---|---|---|---|---|---|---|---|---|---|---|
| (A) Body size | Climate (C) | 1,8 | **7.42*** | 2.52 | **5.52*** | **5.21*** | 0.28 | 2.26 | 3.24 | 1.52 | **4.81*** | 0.42 | 1.89 | 0.16 |
| | Land use (L) | 4,32 | 1.34 | 1.32 | 0.65 | 2.13 | 0.52 | 0.49 | 1.21 | 1.79 | 0.44 | 0.65 | 3.55 | 0.47 |
| | C × L | 4,32 | 0.65 | 2.09 | 1.85 | 1.11 | 0.96 | 0.62 | 1.66 | 1.45 | 1.6 | 2.48 | 16.76 | 0.41 |
| | Season (D) | 2,16 | **63.87*** | **52.13*** | **83.35*** | 1.15 | 0.42 | **4.74** | **13.69*** | 1.2 | **7.55** | 3.25 | 0.11 | **4.87*** |
| | C × D | 2,16 | 0.54 | 0.27 | 1.31 | 1.92 | - | 4 | 3.9 | 0.13 | 0.18 | 0.15 | 6.59 | 0.19 |
| | L × D | 8,64 | 2.44 | 0.8 | 1.81 | 1.3 | 0.18 | 2.22 | 1.5 | **3.33** | 0.86 | 1.97 | 2 | 0.91 |
| | C × L × D | 8,64 | 0.37 | 0.81 | 0.95 | 0.76 | - | 1.1 | 1.36 | 0 | 2.1 | 2.49 | - | 0.37 |
| (B) Density | Climate (C) | 1,8 | **4.78†** | 1.85 | 0.48 | 0.02 | 1.93 | **6.02*** | **5.03†** | **5.03†** | 0.01 | 0 | 0.12 | 1.44 |
| | Land use (L) | 4,32 | **9.47*** | **10.32*** | **5.86** | **5.49** | **3.76*** | 0.74 | 0.81 | **3.68*** | **4.07** | **3.4*** | 1.45 | 0.37 |
| | C × L | 4,32 | 1.1 | 1.13 | 1.1 | 0.51 | 0.31 | 1.05 | 0.17 | 0.88 | 0.35 | 0.68 | 0.24 | 0.36 |
| | Season (D) | 2,16 | **9.45** | **13.28*** | **14.44*** | **6.04*** | **97.61*** | 2.99 | **9.51** | 22.96 | **14.85*** | **5.88*** | **4.4*** | **8.63** |
| | C × D | 2,16 | 0.22 | 0.64 | 0.93 | 0.36 | 3.1 | 0.07 | 3.35 | 2.14 | 0.57 | 0.97 | 0.25 | 0.31 |
| | L × D | 8,64 | 2.02 | 1.05 | 1.53 | 0.86 | **3.24** | **3.71** | 1.89 | **2.37*** | **4.3*** | **2.67*** | 2.02 | 1.84 |
| | C × L × D | 8,64 | 0.37 | 0.56 | 0.59 | 0.77 | 0.81 | 0.45 | 0.71 | 1.35 | 0.67 | 1.39 | 0.78 | 0.84 |
| (C) Biomass | Climate (C) | 1,8 | **8.69*** | **7.15*** | **4.04†** | **6.76*** | 1.67 | **8.86*** | **7.89*** | 0.18 | 0.23 | 0.1 | 1.77 | 0.04 |
| | Land use (L) | 4,32 | **4.73** | **4.24** | **4.91** | 1.94 | 0.84 | 1.55 | 0.51 | 0.55 | 0.47 | 0.5 | 0.78 | 0.65 |
| | C × L | 4,32 | 1.33 | 1.49 | 1.32 | 0.65 | 0.33 | 0.61 | 0.43 | 1.06 | 0.25 | 0.38 | 3.43 | 0.46 |
| | Season (D) | 2,16 | 3.57 | **7.27** | **5.49*** | **6.82** | 4.04 | **13.7*** | **13.58*** | 0.34 | 0.53 | 1.57 | 1.31 | **5.85*** |
| | C × D | 2,16 | 0.9 | 1.34 | 0.28 | 2.05 | - | 3.18 | 3.65 | 0.14 | - | 0.31 | 6.37 | 0.64 |
| | L × D | 8,64 | 2.62 | 1.74 | 1.93 | 1.67 | 0.52 | **3.54** | 1.9 | 2.08 | - | 0.33 | **25.78*** | 1.17 |
| | C × L × D | 8,64 | 0.61 | 0.85 | 0.43 | 1.33 | - | 1.29 | 1 | 1.44 | - | 0.59 | - | 0.51 |

The online version of this article includes the following source data for Table 1:

Source data 1. Density dataset of microarthropods.

Source data 2. Density dataset of microarthropods.

Source data 3. Biomass dataset of microarthropods.

## Intensive land use reduces the density of soil microarthropods

Land-use treatments significantly decreased the density of microarthropods by ~47% from the extensively-used meadow to conventional farming (*Figure 2A*; *Table 1B*). More specifically, these significant land-use effects on microarthropod density were due to the differences between the two main land-use types (grasslands > croplands), and the two grassland types (meadows < pastures) but were not due to differences between land-use management intensities within croplands (i.e., conventional farming and organic farming), and the meadows (i.e., intensively-used meadows and extensively-used meadows), respectively (*Supplementary file 1A*).

Among microarthropod communities, Acari (including its orders, i.e., Oribatida, Mesostigmata, Prostigmata) and some Collembola taxa (i.e., Entomobryidae, Sminthuridae, Katiannidae) significantly decreased in their density in response to land-use intensification (*Figure 2B–H*; *Table 1B*). Similar to total microarthropod densities, these significant land-use effects were mainly caused by the differences between the two main land-use types (with higher densities in grasslands than in croplands) but were less due to the differences between land-use management intensities within the croplands (conventional farming and organic farming), grasslands (meadows and pastures), and meadows (intensively-used meadows and extensively-used meadows), respectively (*Supplementary file 1A*).

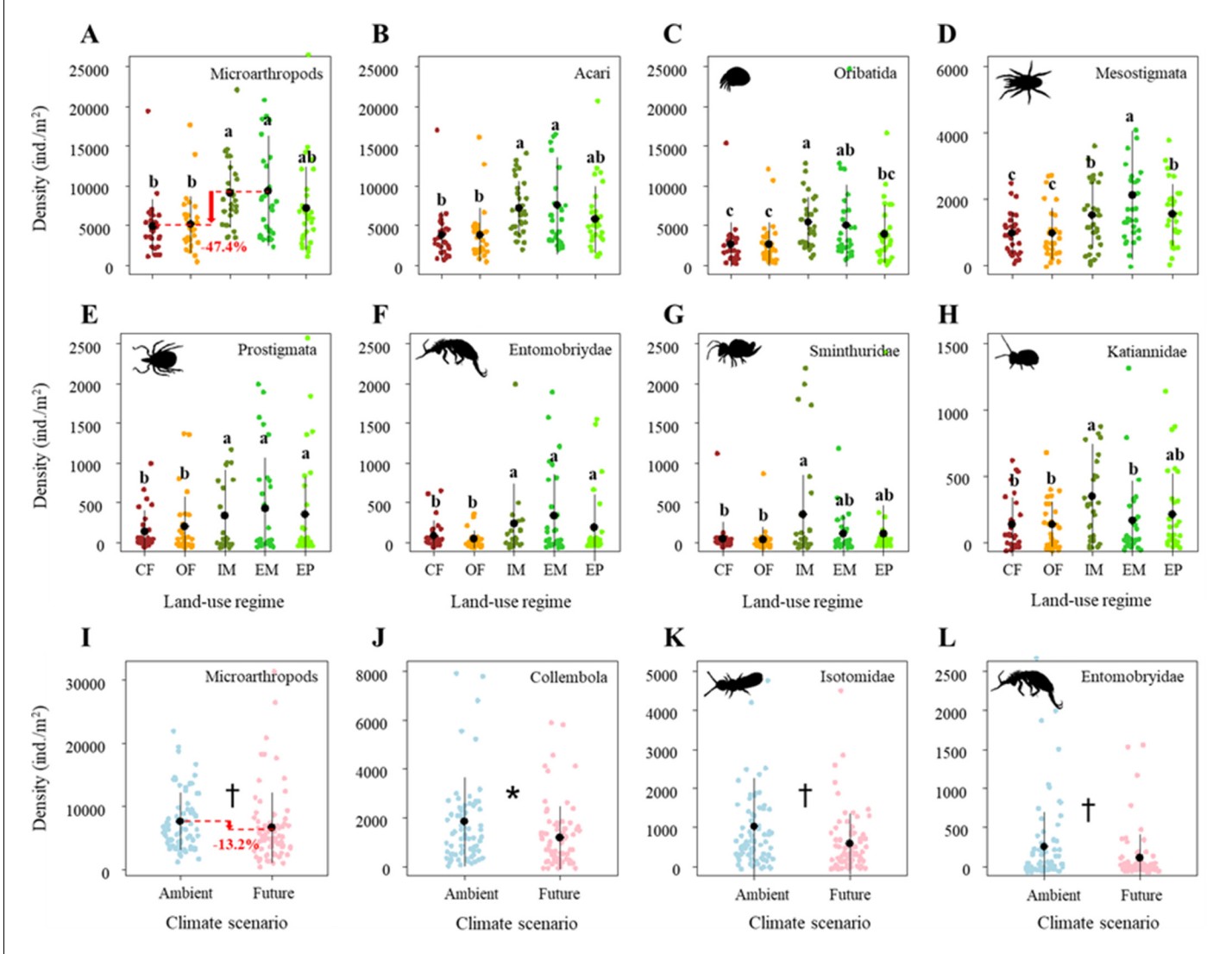

**Figure 2.** Effects of land use on the density of (A) Microarthropods, (B) Acari, (C) Oribatida, (D) Mesostigmata, (E) Prostigmata, (F) Entomobryidae, (G) Sminthuridae, and (H) Katiannidae. The predicted mean ± SD of density for the conventional farming (CF, with all raw data as brown points), organic farming (OF, with all raw data as orange points), and intensively-used meadow (IM, with all raw data as olive drab points), extensively-used meadow (EM, with all raw data as lime green points), and extensively-used pasture (EP, with all raw data as lawn green points). Different lowercase letters denote significant (P < 0.05) differences among land-use regimes based on post-hoc Tukey's HSD tests. Effects of climate on the density of (I) Microarthropods, (J) Collembola, (K) Isotomidae, and (L) Entomobryidae. The predicted mean ± SD of body size for the ambient climate scenario (with all raw data as blue points), and future climate scenario (with all raw data as red points). * and † denote significant (P < 0.05) and marginal (P < 0.10) differences between climate scenarios based on post-hoc Tukey's HSD tests, respectively.

Additionally, climate change effects on microarthropod density were negligible. Specifically, future climate did not significantly affect the density of Acari and most Collembola taxa (*Table 1B*). It only marginally decreased the density of microarthropods, which was mainly caused by a decrease in the density of Collembola (significant), Isotomidae, and Entomobryidae (both marginally significant) (*Figure 2I–L*).

## Climate change and intensive land use reduce the biomass of soil microarthropods

The total biomass of microarthropods was significantly affected by both global change drivers (*Table 1C*). Specifically, climate change significantly reduced the biomass of microarthropods

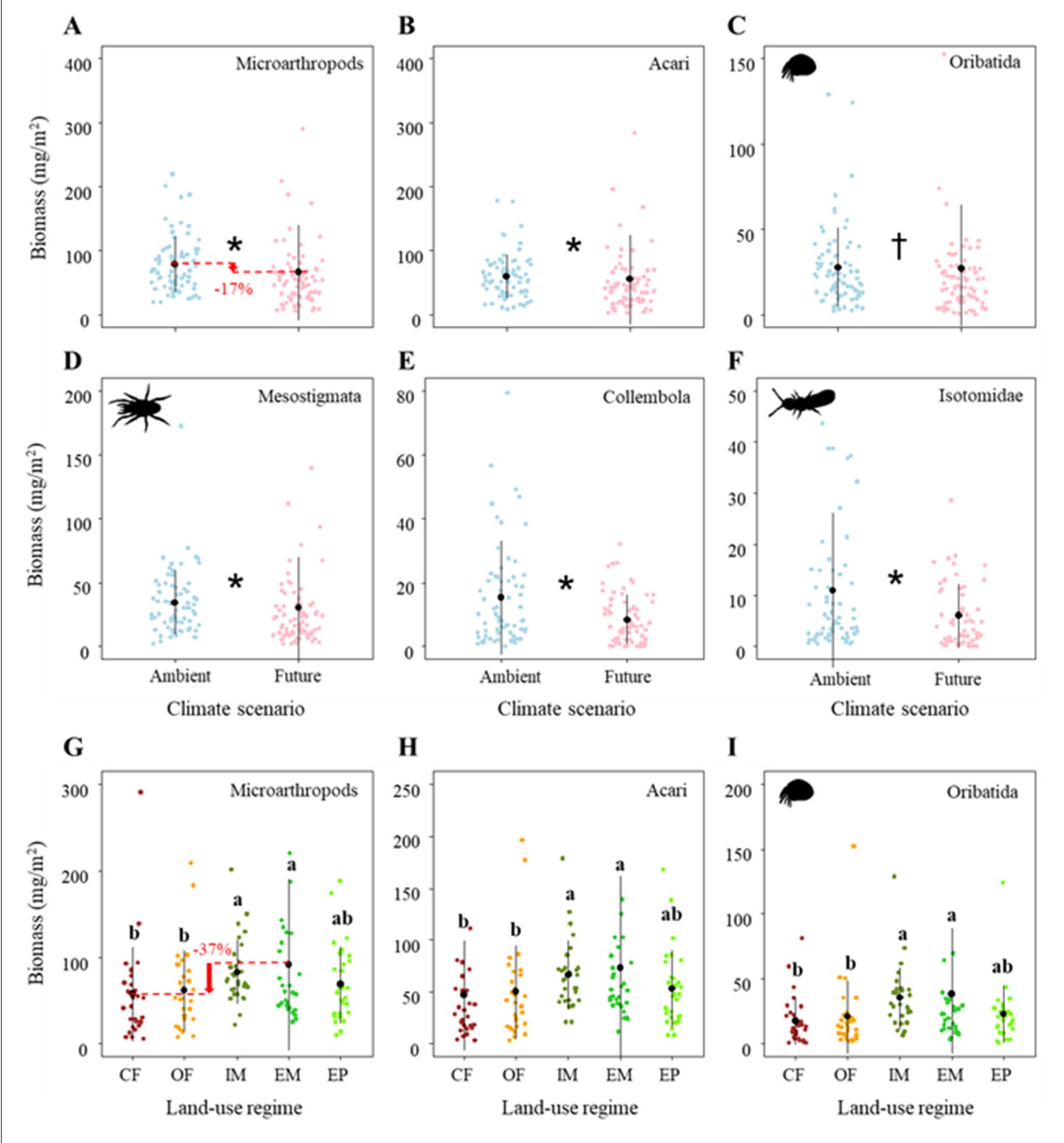

**Figure 3.** Effects of climate on the biomass of (**A**) Microarthropods, (**B**) Acari, (**C**) Oribatida, (**D**) Mesostigmata, (**E**) Collembola, and (**F**) Isotomidae. The predicted mean ± SD of body size for the ambient climate scenario (with all raw data as blue points), and future climate scenario (with all raw data as red points). * and † denote significant ($P < 0.05$) and marginal ($P < 0.10$) differences between climate scenarios based on post-hoc Tukey's HSD tests, respectively. Effects of land use on the biomass of (**G**) Microarthropods, (**H**) Acari, and (**I**) Oribatida. The predicted mean ± SD of density for the conventional farming (CF, with all raw data as brown points), organic farming (OF, with all raw data as orange points), and intensively-used meadow (IM, with all raw data as olive drab points), extensively-used meadow (EM, with all raw data as lime green points), and extensively-used pasture (EP, with all

*Figure 3 continued on next page*

*Figure 3 continued*

raw data as lawn green points). Different lowercase letters denote significant (P < 0.05) differences among land-use regimes based on post-hoc Tukey's HSD tests.

by ~17% (*Figure 3A*; *Table 1C*), which was driven by both Acari (mainly due to its dominant taxa: Oribatida and Mesostigmata) and Collembola (mainly due to its dominant taxon: Isotomidae) (*Figure 3B–F*; *Table 1C*).

Additionally, the total biomass of microarthropods sharply decreased by ~37% from the extensively-used meadow to conventional farming (*Figure 3G*; *Table 1C*), which was driven by a decreased biomass of Acari (mainly due to its dominant taxon: Oribatida) (*Figure 3H–I*; *Table 1C*). These significant land-use effects on the biomass of microarthropods (mostly that of Acari and Oribatida) were consistently driven by the differences between the two main land-use types (i.e., higher biomass in grasslands than in croplands) but not by the different management intensities within grasslands and croplands, respectively (*Supplementary file 1B*).

## Pathways of biomass decrease in soil microarthropods

SEM results further confirmed that climate change and intensive land use reduced the biomass of soil microarthropods indirectly via two different pathways. While biomass loss caused by future climate was mediated by reduced body size, biomass loss caused by intensive land use was mediated by decreased density (*Figure 4*). Besides, in this model, we found that the mean body size of soil microarthropods was negatively correlated with their density (*Figure 4*).

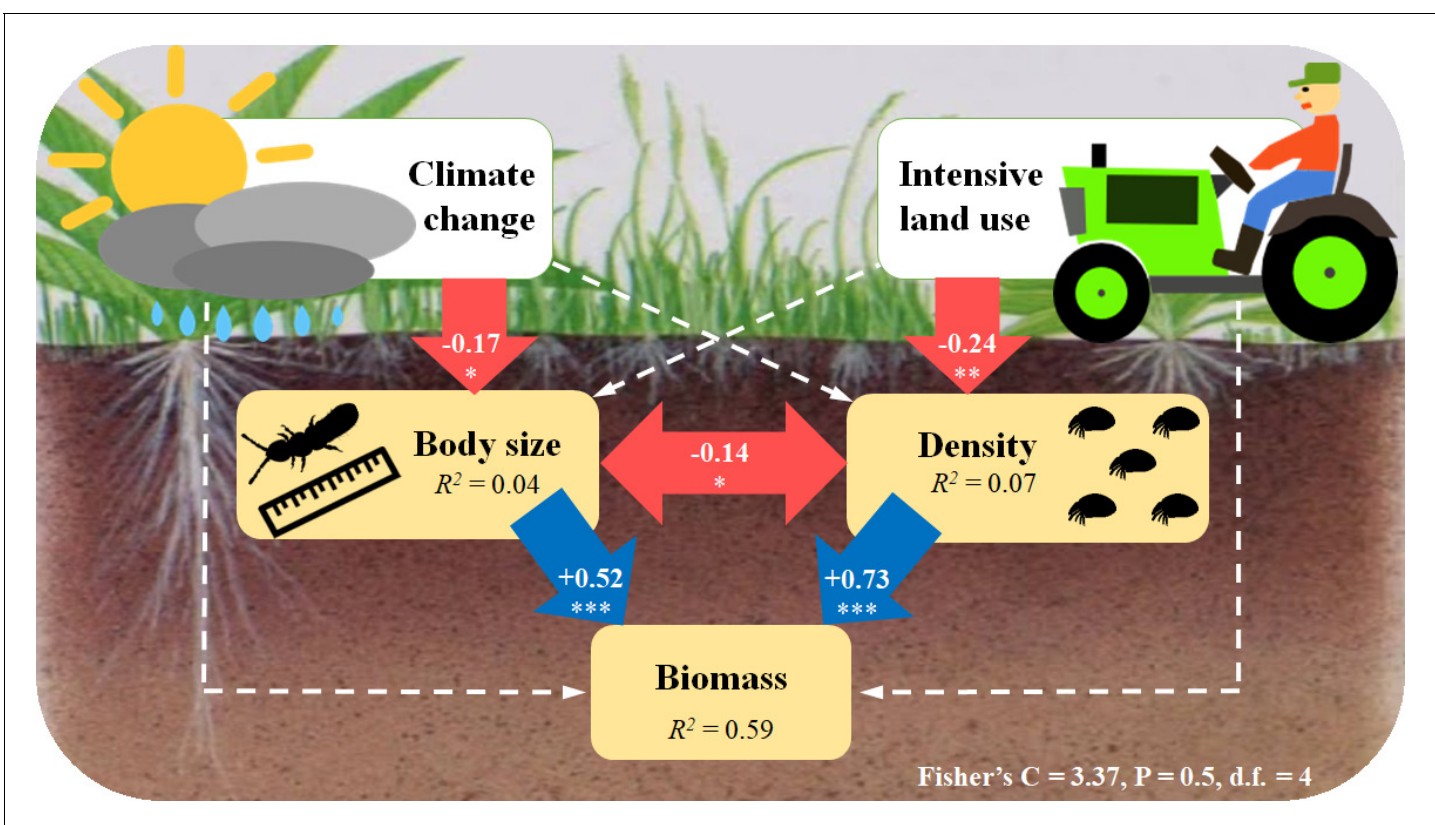

**Figure 4.** Structural equation model (SEM) showing the pathways through which climate change and intensive land use influence soil microarthropod biomass. The final model (AIC = 33.37) is the best-fitting model, with Fisher's C = 3.37; P = 0.5; d.f. = 4. Numbers in the arrows are standardized path coefficients. The blue (positive) and red (negative) one-way arrows indicate significant effects with * = P < 0.05, ** = P < 0.01, *** = P < 0.001. The dashed arrows indicate non-significant effects (P > 0.05) that are still remaining in this model. The double-headed red arrow indicates a significant correlation, with * = P < 0.05. The variance explained ($R^2$) is shown in each panel.

## No interactive effects of climate and land use on soil microarthropods

Contrary to our expectation, there were no significant interactive effects of climate and land use on body size, density, or biomass of soil microarthropods; neither on Acari nor Collembola (*Table 1A-C*). Besides, the full model (overfit, with Fisher's *C* = 0; p-value=1; d.f. = 0) with inclusion of the individual and interactive effects of climate and land use also certified no significant interaction effects on our response variables.

## Seasonal patterns of microarthropod body size, density, and biomass

The body size of Acari was significantly larger in spring than autumn, but no significant effects of season were detected on the body size of Collembola and total microarthropods (*Supplementary file 2A*). The density of microarthropods (including both Acari and Collembola) was significantly higher in autumn than in spring (*Supplementary file 2B*). The biomass of Collembola was significantly higher in autumn than in spring, but no significant season effects were detected on the biomass of Acari and total microarthropods (*Supplementary file 2C*).

## Discussion

The main findings of the present study are as follows: (1) negative, but largely independent, effects of climate change and intensive land use were found on soil microarthropod biomass; and (2) these independent effects can be explained by two dissimilar pathways: climate change reduced microarthropod mean body size and intensive land use decreased microarthropod density. As climate change and intensive land use operated via dissimilar pathways, our hypothesis of synergistic environmental change effects was not supported, indicating that the underlying pathways of climate change effects are consistent across land-use regimes and vice versa.

### Climate change and land-use intensification reduce total microarthropod biomass

In our study, climate change caused a significant reduction of total microarthropod biomass in the soil. Our results are in line with those of *Vestergård et al., 2015*, who demonstrated that the biomass of microarthropods was dramatically reduced by drought, especially when combined with warming. Accordingly, climate warming was shown to exacerbate the drying of soil and thereby the negative drought effects on soil microarthropods (*Thakur et al., 2018*).

For the soil system, we could show with our study that the future climate treatment simulated by increased air and soil temperatures (+0.6°C) and altered precipitation (−20% in summer and +10% in spring/autumn) consistently reduced the body size and total biomass of soil microarthropods across different land-use types and management. This adds to the existing body of literature reporting similar effects for other groups of organisms (*Daufresne et al., 2009*; *Gardner et al., 2011*; *Sheridan and Bickford, 2011*; *Yom-Tov, 2001*) and environmental contexts, thus underlining its validity.

Reduction in body size is supposed to be a universal response of animals to climate change, which is supported by the ecological rules dealing with temperature–size relationships, i.e., Bergmann's rule (*Bergmann, 1848*), James' rule (*James, 1970*), and Temperature–size rule (*Atkinson, 1994*), stating that warmer conditions would lead to organisms with smaller body size (*Gardner et al., 2011*). This phenotypical variation is a widespread pattern across taxa, but there are genetic differences between species (*Forster et al., 2012*). As a consequence, climate change only significantly decreased the body size of some specific taxa (i.e., Oribatida, Mesostigmata, and Sminthuridae) in the present study.

Given the tight connection between ecological properties (e.g., longevity, fecundity, and mortality rates, as well as competitive interactions) and body size (*Chown and Gaston, 2010*; *Savage et al., 2004*; *Thakur et al., 2017*), reductions in mean body size result in changes in community biomass acquisition, and thereby the functioning of ecosystems. For example, the decreased total biomass of soil microarthropods may decelerate the processes of litter decomposition and nutrient cycling, which may also reduce nutrient mineralization processes, plant-available nutrients, and aboveground production.

Intensive land use also reduced the total biomass of microarthropods, but this negative effect was due to lower densities in croplands than in grasslands. This is in accordance with other studies reporting negative effects of intensive land use (*Baker, 1998*; *Birkhofer et al., 2017*) on the density of soil fauna, whereas the conversion from croplands to grasslands was shown to have positive effects (*Zaitsev et al., 2006*). Compared with croplands, grasslands (with less intensive disturbance but more diverse plant communities) provide an environment with more habitats and more accessible food sources, can maintain higher densities of soil organisms (*Alvarez et al., 2001*; *Nyawira Muchane, 2012*; *Scherber et al., 2010*). Accordingly, we found significantly higher levels of microarthropod density and biomass in grasslands than in croplands, which is supported by previous studies (e.g., *de Groot et al., 2016*). Similarly, microarthropod density and biomass are expected to be also reduced by the intensified management practices, e.g., monoculture, heavy use of mineral fertilizers and pesticides in croplands, and frequent fertilization, mowing, and grazing in grasslands; however, we did not find any evidence for this in the present study. This may be explained by opposing effects of such management practices, as the described detrimental impacts on soil communities can be compensated by positive effects, such as elevated plant productivity due to mineral nutrient addition. Likewise, in grasslands, the negative effects of mowing or sheep grazing may be partly compensated by the positive effects of enhanced plant productivity in response to fertilizer usage and sheep manure (*Epelde et al., 2017*). Besides, these management practices appear to have less pronounced impacts on belowground communities than on aboveground communities, and less significant impacts on the abundance or biomass of soil biota than on their biodiversity (*Flohre et al., 2011*; *Tuck et al., 2014*). Therefore, long-term studies are needed to explore if management intensity effects will change over time.

## Extensive land use has limited potential to mitigate the consequences of climate change

It is often suggested that extensive land use may effectively mitigate climate change effects due to higher (aboveground and belowground) diversity and lower anthropogenic disturbance (*Isbell et al., 2017*; *Oliver et al., 2016*). Accordingly, we hypothesized the negative effects of climate change on soil microarthropods could be particularly strong in intensive land use, whereas they could be compensated by extensive land use. In contrast to this hypothesis, we did not observe any significant interaction effects of climate and land use on body size, density, and total biomass of soil microarthropods. These results showed that the effects of climate change on soil microarthropods were consistent across different land-use regimes, suggesting that negative climate change effects will not be exacerbated by intensive land use, nor mitigated by extensive land use. This finding calls for novel management strategies to alleviate the consequences of climate change. Our study provides the first mechanistic insights into the underlying pathways of changes in soil communities that may inform such novel management approaches.

## Climate change and land-use intensification decrease soil microarthropod biomass via independent pathways

Climate change and intensive land use decreased microarthropod biomass via indirect and independent pathways, namely, climate change reduced the mean body size and intensive land use decreased overall densities. This is the first empirical evidence for such contrasting pathways underlying different environmental change factors in a full-factorial experiment. Consistent climate change and land-use effects under different land-use regimes and climate contexts, respectively, suggest that (1) the identified pathways may apply to a wide range of environmental conditions, and (2) current extensive land-use regimes do not mitigate negative climate change effects on ecosystems. However, we do not know yet if the outcomes are due to pure assembly mechanisms or if evolution also plays an important role. Besides, the results of the SEM also revealed that the mean body size of soil microarthropods was negatively correlated with their population density, which is supported by a hypothesis of allometric (e.g., body size-abundance) relationships (*Comor et al., 2014*; *Mulder et al., 2011*; *Niu et al., 2015*), but this observation needs further exploration.

Additionally, body size-mediated effects of climate change on soil microarthropod communities may have profound implications for total community composition and ecosystem processes driven by soil organisms, such as decreased litter decomposition rates (*Taylor et al., 2010*). We, therefore,

encourage future studies to investigate how microarthropod biomass shifts affect soil ecological processes (like litter decomposition and nutrient dynamics) and food web relations in the context of climate change. Moreover, future studies should investigate if decreasing mean body size of soil microarthropods in response to climate change is due to species turnover toward smaller *r*-selected species, shrinking body size within species, or both. By contrast, the reduction in microarthropod densities in soil due to land-use intensification was not accompanied by reductions in mean body size. Future studies should, therefore, explore which other traits of soil microarthropods are influenced by climate change and land-use intensity and how they link to ecosystem functioning.

## Materials and methods

### Study site

The Global Change Experimental Facility (GCEF) is a large field research platform of the Helmholtz-Center for Environmental Research (UFZ), which is located in Bad Lauchstädt, Germany (51˚ 23′ 30N, 11˚ 52′ 49E, 116 m a.s.l.) and was established on a former arable field with the last cultivation in 2012. This arable field is characterized by a low mean annual precipitation of 498 mm and a mean temperature of 8.9˚C. The soil is a Haplic Chernozem with neutral pH (5.8–7.5), high nutrient contents (i.e., total carbon and total nitrogen varied between 1.71–2.09% and 0.15–0.18%, respectively), and humus content of 2% reaching down to a depth >40 cm. The soil is known for its high water storage capacity (31.2%) and density (1.35 g/cm$^3$), ensuring a relatively low sensitivity to drought stress (*Altermann et al., 2005*; *Iwg WRB, 2007*).

### Experimental set-up

The GCEF platform was designed to investigate the effects of future climatic conditions on ecosystem processes under different land-use regimes (*Schädler et al., 2019*). Each of the 10 main plots was divided into 5 subplots (each 16 m x 24 m), resulting in 50 subplots in total. The five subplots within each main-plot were randomly assigned to one of the five land-use regimes: (1) conventional farming (CF; cropland), (2) organic farming (OF; cropland), (3) intensively-used meadow (IM; grassland), (4) extensively-used meadow (EM; grassland), and (5) extensively-used pasture (EP; grassland). Half of the main plots are subjected to an ambient climate scenario, the other half to a future climate scenario (*Figure 5B,C*; *Supplementary file 3*). For our first hypothesis, we not only assessed the general land-use effects, but also further contrasted the effects of land-use types (croplands vs. grasslands), croplands (conventional farming vs. organic farming), grasslands (meadows vs. pastures), and meadows (intensive meadows vs. extensive meadows).

Croplands and intensive meadows were established on the respective subplots in the summer and autumn of 2013. The intensive meadow is a conventionally used mixture of forage grasses (20% *Lolium perenne*, 50% *Festulolium*, 20% *Dactylis glomerata*, and 10% *Poa pratensis*). Within the study period, winter wheat (2015) and winter barley (2016) were grown in these two croplands (the detailed crop rotational sequences are shown in *Supplementary file 3*). In extensively-used meadow and pasture, we repeatedly sowed target plant seeds (legumes, grasses, and non-leguminous dicots) during the spring and autumn of 2014. See *Supplementary file 3*, and *Schädler et al., 2019* for the detailed description of land-use regimes.

The climate treatment is based on a consensus scenario for Central Germany in the period from 2070 to 2100, which was derived from 12 climate simulations based on four different emission scenarios using three established regional climate models: COSMO-CLM (*Rockel et al., 2008*), REMO (*Jacob and Podzun, 1997*), and RCAO (*Döscher et al., 2002*). The consensus scenario predicts an increase of mean temperature across all seasons by ~1˚C. For precipitation, the mean values of the 12 projections resulted in an experimental treatment consisting of a ~ 9% increase in spring (March–May) and autumn (September–November) and a ~ 21% decrease in summer (June–August).

All main plots are equipped with steel framework elements (5.50 m height) to account for possible side effects of the construction itself. Main plots that are subject to future climate are further equipped with mobile shelters, side panels, and rain sensors to allow for alterations in precipitation amounts. Shelters were automatically closed from sundown to sunrise to increase night temperature ~0.6˚C (*Schädler et al., 2019*). The night closing during these periods increased the mean daily air temperature at 5 cm height by 0.55˚C, as well as the mean daily soil temperature in 1

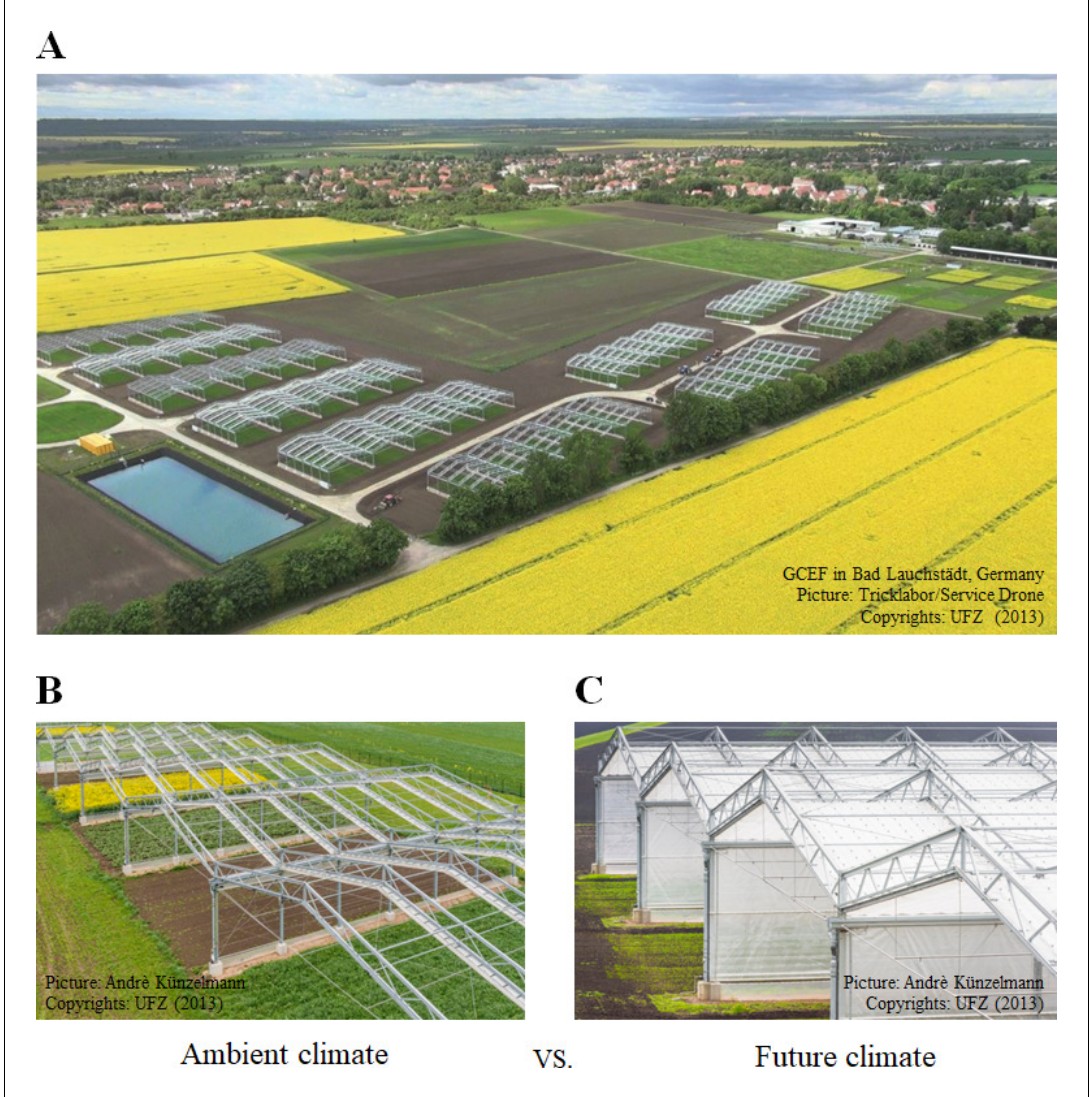

**Figure 5.** Global Change Experimental Facility (GCEF). (**A**) Aerial image of the experimental set-up of the GCEF in Bad Lauchstädt, Germany. Climate treatments as the main plot factor with two levels: (**B**) Ambient climate vs. (**C**) future climate.

cm and 15 cm depth by 0.62°C and 0.50°C, respectively. Using an irrigation system, we added rain water to achieve ~110% of ambient rainfall to the main plots with the future climate in spring and autumn. Additionally, the rain sensors associated with the irrigation system were used to regulate precipitation on the future climate main plots to ~80% of ambient rainfall in summer. As a result, precipitation was increased by 9.2% to 13.6% in spring and autumn, and decreased by 19.7% to 21.0% in summer in both years, respectively. Climate manipulation started in spring 2014. During our experiment, the roofs were active in 2015 (from 15th February to 11th December) and 2016 ( from 22nd March to 29th November).

## Assessment of soil microarthropods

Soil samples were collected three times (autumn 2015, spring 2016, and autumn 2016) during a 1.5 year study period. At each sampling point, three soil cores (Ø 6 cm, 5 cm depth) were taken per sub-plot to extract microarthropods (Collembola and Acari) using a Macfadyen high-gradient extraction method (*Macfadyen, 1961*). Using a digital microscope (VHX-600, Keyence Corp., Osaka, Japan), Acari were identified to the order level, that is, Oribatida, Mesostigmata, and Prostigmata; and

Collembola were identified to the family level, that is, Isotomidea, Entomobryidae, Katiannidae, Sminthuridae, Hypogastruridae, and Onychiuridae. For all taxa, we counted the number of individuals, and measured the body size (length, μm) of each individual using the measurement function of the VHX microscope.

## Statistical analysis

The biomass (M, μg) of microarthropod groups (Order level: Collembola, Oribatida, Mesostigmata. and Prostigmata) was calculated according to a specific formula: Log M = a + b × log L with L as the body size (length) of microarthropods (μm), with Collembola: a = −1.8479; b = 2.3002; Mesostigmata: a = 2.064; b = 2.857; Oribatida: a = 2.117; b = 2.711; Prostigmata: a = 2.124; b = 2.808 (*Ganihar, 1997*; *Mercer et al., 2001*). For each subplot, the mean body size, population density, and total biomass of microarthropods, Acari and its orders (Oribatida, Mesostigmata, and Prostigmata), Collembola and its families (Isotomidea, Entomobryidae, Katiannidae, Sminthuridae, Hypogastruridae, and Onychiuridae) were represented by mean ± SD and analyzed.

We analyzed the effects of climate (two levels: ambient vs. future), land use (five categories: OF, CF, IM, EM, EP), season (three times, with two in autumn and one in spring), and their interactions on the response variables using generalized linear mixed models with sampling season as repeated factor (in SAS v 9.4). Body size and biomass data were log-transformed prior to analyses to meet the requirements of parametric statistical tests. Count data (abundance/density) were analyzed assuming Poisson-distributed residuals with a log-link function. In a few cases, there was indication of overdispersion according to generalized Chi-squared/d.f. ratios, and therefore we assumed negative binomial-distributed residuals. The superior fit of the selected distributions was further confirmed using the Akaike Information Criterion.

If the land-use effects were significant on specific response variables, then we further made orthogonal contrasts to assess the effects of land-use types (croplands vs. grasslands), cropland management intensity (conventional farming vs. organic farming), grassland type (meadows vs. pastures), and meadow management intensity (intensive meadows vs. extensive meadows), respectively. In addition, we had an unbalanced dataset with two autumn samplings, but one spring sampling; therefore, we also ran a contrast (autumn vs. spring) to explore if there is a season effect on the body size, density, and biomass of total soil microarthropods, Acari, and Collembola.

Furthermore, we ran path models using the 'piecewiseSEM' package (*Lefcheck, 2016*) to disentangle the potential causal direct and indirect pathways by which climate, land use, and their interaction influence the response variables. More precisely, (1) the models were created using generalized linear mixed models in a full model with inclusion of the individual and interactive effects of climate change (i.e., ambient → future) and land-use intensification (i.e., following an increased intensity gradient: EM → EP → IM → OF → CF) on the body size, density, and biomass of soil microarthropods. According to Shipley's test of d-separation yielding the Fisher's C statistic (Chi-square distributed; *Shipley, 2009*), our full model was saturated (overfit, with Fisher's C = 0; P = 1; d.f. = 0) and no interactive effects of climate and land use were detected. Therefore, we subsequently reduced the number of pathways (i.e., climate × land use interaction) to meet the criteria of Shipley's test of d-separation in the reduced (final) model. We only reported the standardized coefficient for paths of this final model in the results. (2) We tested if climate change and land-use intensification affected soil microarthropod biomass (main response variable) via reductions in body size or density, and if the two global change drivers differ in their pathways.

## Acknowledgements

The first author Rui Yin appreciates the funding by the Chinese Scholarship Council (CSC) (File No.201406910015). All authors appreciate the Helmholtz Association, Federal Ministry of Education and Research, the State Ministry of Science and Economy of Saxony-Anhalt and the State Ministry for Higher Education, Research and the Arts Saxony to fund the Global Change Experimental Facility (GCEF) project. This project also received support from the European Research Council (ERC) under the European Union's Horizon 2020 research and innovation program (grant agreement No. 677232 to NE). Further support came from the German Centre for Integrative Biodiversity Research (iDiv) Halle-Jena-Leipzig, funded by the DFG (FZT 118). We also appreciate the staff of the Bad Lauchstädt Experimental Research Station (especially Ines Merbach and Konrad Kirsch) for their hard work in

maintaining the plots and infrastructures of the GCEF, and Dr. Stefan Klotz, Prof. Dr. Francois Buscot and Dr. Thomas Reitz for their roles in setting up the GCEF. In addition, particularly thank Prof. Dr. Paul Kardol for his polishing of the manuscript and his valuable comments.

# Additional information

## Funding

| Funder | Grant reference number | Author |
|--------|------------------------|--------|
| H2020 European Research Council | No. 677232 | Nico Eisenhauer |
| Deutsche Forschungsgemeinschaft | FZT 118 | Nico Eisenhauer |
| China Scholarship Council | 201406910015 | Rui Yin |

The funders had roles in study design, data collection and interpretation, or the decision to submit the work for publication.

## Author contributions

Rui Yin, Conceptualization, Data curation, Software, Formal analysis, Investigation, Methodology, Writing - original draft; Julia Siebert, Software, Writing - original draft; Nico Eisenhauer, Conceptualization, Supervision, Funding acquisition, Methodology, Writing - original draft, Project administration, Writing - review and editing; Martin Schädler, Conceptualization, Data curation, Supervision, Funding acquisition, Investigation, Methodology, Writing - original draft, Project administration, Writing - review and editing

## Author ORCIDs

Rui Yin (iD) https://orcid.org/0000-0002-4580-1317
Nico Eisenhauer (iD) https://orcid.org/0000-0002-0371-6720

## Decision letter and Author response

Decision letter https://doi.org/10.7554/eLife.54749.sa1
Author response https://doi.org/10.7554/eLife.54749.sa2

# Additional files

## Supplementary files

• Supplementary file 1. Results from generalized linear mixed models with linear contrasts testing the effects of land-use management intensity within land-use types (croplands vs. grasslands), croplands (conventional farming vs. organic farming), grasslands (meadows vs. pastures), and meadows (intensive meadows vs. extensive meadows), respectively, on (A) density of microarthropods, Acari, Oribatida (Orib_), Mesostigmata (Meso_), Prostigmata (Pros_), Entomobryidae (Ento_), Katiannidae (Kati_), Sminthuridae (Smin_), and (B) biomass of microarthropods, Acari, Oribatida (Orib_). F-values are given and the significant effects are in bold font, with †= $P < 0.1$, * = $P < 0.05$, ** = $P < 0.01$, *** = $P < 0.001$. The predicted mean ± SD are given and different lowercase letters denote significant ($P < 0.05$) differences between land-use management intensities.

• Supplementary file 2. Results from generalized linear mixed models with linear contrasts testing the effects of season (autumn vs. spring) on (A) body size, (B) density, and (C) biomass of microarthropods, Acari, and Collembola. *F*-values are given and the significant effects are in bold font, with ** = $P < 0.01$, *** = $P < 0.001$. The predicted mean ± SD are given and different lowercase letters denote significant ($P < 0.05$) differences between the two seasons.

• Supplementary file 3. Descriptions of five land-use regimes.

• Transparent reporting form

**Data availability**

All data generated or analysed during this study are included in the manuscript and supporting files.

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
