## [Decision Letter]

**Acceptance summary:**

This paper reports results from perhaps the largest experiment worldwide simulating the combination of predicted climate change (increased temperature and altered precipitation) and land-use intensification. Both of these global change drivers independently reduce the biomass of soil microarthropods, the first by reducing their average size and the second by reducing their numbers. These changes may have further negative effects on soil functions.

**Decision letter after peer review:**

Thank you for submitting your article "Climate change and intensive land use reduce soil animal biomass through dissimilar pathways" for consideration by *eLife*. Your article has been reviewed by three peer reviewers, one of whom is a member of our Board of Reviewing Editors, and the evaluation has been overseen by Christian Rutz as the Senior Editor. The following individual involved in the review of your submission has agreed to reveal their identity: Eva Knop (Reviewer #1).

The reviewers have discussed the reviews with one another and the Reviewing Editor has drafted this decision to help you prepare a revised submission. There was some discussion whether your paper should be rejected, but we believe that if you take our concerns seriously, it should be possible to prepare a revision that is suitable for publication.

Summary:

This is a very nice and well written manuscript on how two major global change drivers, namely climate change and land-use change, affect the density and body size of soil organisms, with consequences for biomass. Even though the authors did not find interactive effects of the two global change drivers, they nicely show how the two global change drivers affect soil biomass differently.

Essential revisions:

The authors assembled a very rich dataset. However, the corresponding analyses do not take advantage of this richness as explained in the following sections.

First, their experimental treatments don't map well onto how they are discussed implicitly or explicitly in Introduction and Discussion. The global-change treatment is a combination of a less than predicted warming in the night (and not day?) and a more substantial change in rainfall pattern. Thus, interpretations in terms of responses to warming, even though tempting, should be made less strongly and instead effects of changed rainfall patterns be discussed in the context of other literature as well. The land-use treatments are interpreted as land-use intensity, but the corresponding contrasts are not made in the statistical analysis. Instead a much weaker approach of multiple-comparison tests is used. The authors should change this and make orthogonal contrasts for their land-use treatments. For example, they could address focused 1-degree of freedom (i.e., directional) hypotheses and their interactions with the global-change treatment: C1 = F vs. non-F, C2 = P vs. M for non-F, C3 = CF vs. OF for F, C4 = IM vs. EM for M. In the ANOVA tables, the SS of the current land-use treatment will be partitioned into the corresponding 4 contrasts.

Second, the statistical models should be reformulated to include time and taxa as additional explanatory terms of interest (i.e., in the fixed part of the model). Thus, observation time (repeated measure) should be used as a fixed effect (or at least the autumn vs. spring contrast thereof), and its interactions with the treatment factors also be analyzed. Furthermore, as indicated in our comments on the pre-submission, it would be better to also include taxon as an explanatory term and to prepare a long data table. This has the advantage that the main effects of treatments and their interaction is like a test for an overall size, density or biomass response (e.g., currently the total biomass is not being analyzed). The taxon term can then be split into the three major groups and the subgroups within each of the major groups (all as fixed effects, NOT as random effects!). If necessary, the Reviewing Editor can help the authors to formulate the appropriate statistical models.

The above changes to the statistical analysis are needed in order to reconsider many currently unsupported statements, e.g., about different responses of different groups of animals or about effects of different levels of land-use intensity. Furthermore, the authors should explain their analysis better. For example, they write that they used generalized linear-mixed models but they don't say what error distribution and link function they used and don't mention if they checked residual distributions. Oddly, they used path analysis also for the ANOVAs. In fact, the SEMs do not add anything to the interpretation and it would be better to do proper mixed models as explained above. The reason is that their biomass is a simple linear combination of body size and density, which invalidates a SEM in which all three variables are considered as measured independently or at least not as 100% dependent. The SEM is also not necessary, because all is already said by the individual analyses that the global-change treatment mainly affects body size and biomass and the land-use treatment mainly affects density and biomass.

---

## [Author Response]

Essential revisions:The authors assembled a very rich dataset. However, the corresponding analyses do not take advantage of this richness as explained in the following sections.First, their experimental treatments don't map well onto how they are discussed implicitly or explicitly in Introduction and Discussion. The global-change treatment is a combination of a less than predicted warming in the night (and not day?) and a more substantial change in rainfall pattern. Thus, interpretations in terms of responses to warming, even though tempting, should be made less strongly and instead effects of changed rainfall patterns be discussed in the context of other literature as well. The land-use treatments are interpreted as land-use intensity, but the corresponding contrasts are not made in the statistical analysis. Instead a much weaker approach of multiple-comparison tests is used. The authors should change this and make orthogonal contrasts for their land-use treatments. For example, they could address focused 1-degree of freedom (i.e., directional) hypotheses and their interactions with the global-change treatment: C1 = F vs. non-F, C2 = P vs. M for non-F, C3 = CF vs. OF for F, C4 = IM vs. EM for M. In the ANOVA tables, the SS of the current land-use treatment will be partitioned into the corresponding 4 contrasts.

Thank you for your helpful comments and suggestions. Addressing these points, we:

1) Rewrote some parts of the Introduction and Discussion in order to better map on our experimental design and objectives. The predicted climate change treatment in this study is a combination of passive warming and altered precipitation patterns. In the previous version of the manuscript, we focused more on warming effects and less on the effects of (altered) precipitation. Therefore, in the current version, we are putting more emphasis on the effects of altered precipitation patterns and have added respective references to the text. Moreover, we explained in the general discussion why it is so critical to focus on a realistic climate change scenario (e.g., warming + alter precipitation) for a specific geographical area and presented recent support for our approach (e.g. Korell et al., 2020). Please see our revisions in the third paragraph of the Introduction.

2) Further explained the aspect of the predicted warming effects underlying the treatment. On future climate plots, shelters were automatically closed from sundown to sunrise to increase night temperature by 0.6-1°C above ambient levels. Meanwhile, the night closing during these periods increased the mean daily air temperature at 5 cm-height by 0.55°C, as well as the mean daily soil temperature in 1 cm- and 15 cm-depth by 0.62°C and 0.50°C, respectively (Please see changes in the last paragraph of the subsection “Experimental set-up”). We also refer to the respective design paper that presents a very detailed assessment of the experimental treatments (Schädler et al., 2019).

3) Added the corresponding contrasts to our model to further assess the effects of land-use type (‘croplands vs. grasslands’), cropland management intensity (‘conventional farming vs. organic farming’), grassland type (‘meadows vs. pastures’), and meadow management intensity (‘intensive meadows vs. extensive meadows’). This contrast analysis helped us to better understand the significant effects of land use on our response variables, which were mainly caused by the differences among the land-use types, and/or the intensity of land-use management for a particular land-use type. Thanks again for this constructive suggestion (Please see Supplementary file 1).

Second, the statistical models should be reformulated to include time and taxa as additional explanatory terms of interest (i.e., in the fixed part of the model). Thus, observation time (repeated measure) should be used as a fixed effect (or at least the autumn vs. spring contrast thereof), and its interactions with the treatment factors also be analyzed. Furthermore, as indicated in our comments on the pre-submission, it would be better to also include taxon as an explanatory term and to prepare a long data table. This has the advantage that the main effects of treatments and their interaction is like a test for an overall size, density or biomass response (e.g., currently the total biomass is not being analyzed). The taxon term can then be split into the three major groups and the subgroups within each of the major groups (all as fixed effects, NOT as random effects!). If necessary, the Reviewing Editor can help the authors to formulate the appropriate statistical models.

Thank you for your helpful comments and suggestions. Accordingly:

1) We revised the statistical models to include date as an additional explanatory term of interest (i.e., in the fixed part of the model). More precisely, we specified the factor date in our model including the interactions with climate and land use, e.g. Microarthropods_Biomass = Climate*Landuse*Date, and used the specification “random Date/subject=Plot_ID type=AR(1) residual”, to treat the factor ‘Date’ as a fixed repeated factor with a first-order autoregressive structure. The updated results are shown in Table 1.

2) The revised analyses revealed that “Date” was only weakly modulating the effects of climate and land use on soil microarthropods, but that rather individual date effects had significant effects. Therefore, we continued to further contrast only the effects of date ‘autumn vs. spring’ on total microarthropods and their two main groups (i.e., Acari and Collembola); please see Supplementary file 2.

3) We agree that considering taxa as factors in a new (but very large) model would be an elegant way to statistically assess overall and taxa-specific responses. However, in the new version of the manuscript, we now present detailed responses of the single taxa in tables and figures. We therefore think that this new analysis including the taxa in the fixed effect model would not add much information and – also after the consultation of the editor by the authors – suggest to omit this analysis.

The above changes to the statistical analysis are needed in order to reconsider many currently unsupported statements, e.g., about different responses of different groups of animals or about effects of different levels of land-use intensity. Furthermore, the authors should explain their analysis better. For example, they write that they used generalized linear-mixed models but they don't say what error distribution and link function they used and don't mention if they checked residual distributions. Oddly, they used path analysis also for the ANOVAs. In fact, the SEMs do not add anything to the interpretation and it would be better to do proper mixed models as explained above. The reason is that their biomass is a simple linear combination of body size and density, which invalidates a SEM in which all three variables are considered as measured independently or at least not as 100% dependent. The SEM is also not necessary, because all is already said by the individual analyses that the global-change treatment mainly affects body size and biomass and the land-use treatment mainly affects density and biomass.

Thank you for your constructive comments and suggestions.

1) Actually, in our case body mass is not really a linear combination of body size and density since for different taxa, there are different conversion factors for the body size/biomass-relationship. Given that the SEM combines all of the data according to our initial conceptual model and is able to differentiate different effect pathways in a synthetic way, we believe that this figure will be the most cited piece of our work and will increase its impact. This is why we would like to keep our new SEM results, as they (i) provide a test of the main relationships in a multivariate analysis, and (ii) increase the impact of the paper by providing a text book-like figure on the main results.

2) For keeping it, we have rerun the SEM models with a revised R script. Compared to our old SEM results, we gained additional insights (Figure 4). More specifically, the new SEM results (i) confirmed that climate change and intensive land use reduced the biomass of soil microarthropods via two different paths. While biomass loss caused by future climate was mediated by reduced body size, biomass loss caused by intensive land use was mediated by decreased density. Moreover, these new analyses (ii) showed that the mean body size of soil microarthropods was negatively correlated with their population density. The second finding can be supported by the allometric (e.g., body size-abundance) relationships hypothesis (Mulder et al., 2011; Comor et al., 2014; Niu et al., 2015).

But all in all, we leave the final decision up to the editors, but would recommend to keep the SEM as it will be the main figure of the paper that will be shown in presentations and teaching.

Reference:

Korell, L., Auge, H., Chase, J. M., Harpole, W. S., Knight, T. M. (2020): We need more realistic climate change experiments for understanding ecosystems of the future. Global Change Biology 26(2), 325-327